# Evaluation of Liver Parenchyma in Dogs with Hyperlipidemia Using Ultrasound Attenuation Imaging (ATI)

**DOI:** 10.3390/vetsci11100454

**Published:** 2024-09-25

**Authors:** Tina Pelligra, Simonetta Citi, Veronica Marchetti, Verena Habermaass, Sara Tinalli, Caterina Puccinelli

**Affiliations:** Department of Veterinary Sciences, University of Pisa, Via Livornese Lato Monte, San Piero a Grado, 56122 Pisa, Italy; simonetta.citi@unipi.it (S.C.); veronica.marchetti@unipi.it (V.M.); verena.habermaass@phd.unipi.it (V.H.);

**Keywords:** ultrasound attenuation imaging, ATI, dog, steatosis, hyperlipidemia

## Abstract

**Simple Summary:**

Pathological hyperlipidemia is a significant emerging condition in dogs, associated with liver diseases such as hepatic lipidosis or steatosis. Ultrasound attenuation imaging (ATI) is used in humans to quantitatively evaluate liver parenchyma. The aim of this prospective study was to verify the applicability of ATI on hyperlipidemic dogs. Fifty-three dogs underwent clinical examination, blood tests, abdominal ultrasound and ATI between January 2021 and December 2022, divided into two groups: a total of 21 healthy dogs (A) and 32 hyperlipidemic dogs (B). The dogs of Group B were divided into mild hyperlipidemic (B1; n = 15) and moderate/severe hyperlipidemic (B2; n = 17). Based on the qualitative B-mode evaluation, the grade of severity of hepatic steatosis was classified into four degrees, from normal to severe, with 2, 16, 14 and 2 dogs, respectively. The mean AC value was significantly higher in Group B (0.95 ± 0.23 dB/cm/MHz) compared to Group A (0.81 ± 0.10 dB/cm/MHz). No statistically significant differences in AC values were found between Group B1 and Group B2. In conclusion, ATI could be a useful diagnostic tool in patients suspected of hepatic steatosis.

**Abstract:**

(1) Background: Ultrasound attenuation imaging (ATI) is an innovative technique that allows for the evaluation of the degree of lipid infiltration of the liver parenchyma in a simple and non-invasive way. The objective of this study was to verify the applicability of the ATI method in the evaluation of hyperlipidemia. (2) Methods: This study included 53 dogs between January 2021 and December 2022, of which 21 were healthy (A) and 32 had hyperlipidemia (B). The dogs of Group B were divided into mild hyperlipidemic (B1; n = 15) and moderate/severe hyperlipidemic (B2; n = 17). Each dog underwent biochemical examination, B-mode ultrasound and ATI investigation at the liver level via a right intercostal approach. (3) Results: The mean AC value was significantly higher in Group B (0.95 ± 0.23 dB/cm/MHz) compared to Group A (0.81 ± 0.10 dB/cm/MHz). No statistically significant differences were highlighted regarding the ATI values between the subjects with mild and moderate/severe hyperlipidemia. (4) Conclusion: ATI could be a promising method for the non-invasive evaluation of hepatic steatosis in veterinary medicine.

## 1. Introduction

Hyperlipidemia is a canine pathological condition that has recently attracted increasing interest [1,2]. The term hyperlipidemia refers to an increase in the concentration of lipids (i.e., triglycerides, cholesterol or both) in the blood [2].

This may be the result of a primary, often inherited, defect in plasma lipoprotein metabolism or arise as a secondary consequence of systemic disease [3]. Primary hyperlipidemia is rare in dogs but has been recognized as a familial, breed-related, inherited condition in Miniature Schnauzers, characterized by hypertriglyceridemia. Primary hyperlipidemia has been described in other breeds, such as Beagles, Rough Collies, Shetland Sheepdogs, Doberman Pinchers and Rottweilers [2,3,4].

The most frequent form of hyperlipidemia in dogs is secondary to other pathological causes. The most common are endocrine disorders: hypothyroidism, diabetes mellitus and hyperadrenocorticism [4]. There are also other possible causes of hyperlipidemia in dogs, such as obesity, high-fat diets, protein losing nephropathy, pancreatitis, cholestasis, lymphoma, congestive heart failure due to dilated cardiomyopathy and certain medications [2,3,4].

Although hyperlipidemia itself does not seem to appear to be the direct cause of major clinical signs, in some cases, clinically important and potentially life-threatening secondary diseases may develop, such as hepatobiliary disease characterized by hepatocellular accumulation of lipids within hepatocytes, often referred to as hepatic lipidosis or steatosis [2,3,4,5].

In human medicine, steatotic liver disease (SLD) is one of the most common causes of chronic liver disease, and it is characterized by the accumulation of fat in liver parenchyma. Steatotic liver disease includes alcohol-associated liver disease (ALD) and metabolic dysfunction associated steatotic liver disease (MASLD), which can evolve into metabolic dysfunction associated steatohepatitis (MASH) [6,7]. According to older nomenclature, MASLD and MASH are also frequently reported as NAFL (non-alcoholic fatty liver) and non-alcoholic steatohepatitis (NASH), respectively.

Several studies on NAFLD have demonstrated the close correlation in the pathological pathway between simple steatosis and fibrosis, through the intermediate inflammatory stage of NASH to cirrhosis [8].

The quantification of hepatic steatosis is important since it provides information about the severity of the disease [9]. In human medicine, a liver biopsy is the gold standard for the diagnosis and staging of fatty liver disease; however, magnetic resonance imaging proton density fat fraction (MRI PDFF), a non-invasive fatty liver quantitative imaging technique, is frequently used [10,11,12]. Ultrasound is the most commonly used imaging modality to assess the presence and severity of fatty liver as it is safe, widely available and causes little discomfort to the patient [9,10,12]. The ultrasound technique used exploits the principle according to which the fatty liver attenuates the ultrasound beams [12,13]. On this basis, the degree of hepatic steatosis is usually graded using a four-point scale of echogenicity: normal (grade 0), mild (grade 1), moderate (grade 2) and severe (grade 3) [13,14]. However, grayscale ultrasound has disadvantages, such as the subjective nature of the examination, low accuracy for mild steatosis and intra/interobserver variability [15]. In recent years, new ultrasound methods have been developed to quantitatively assess the pathological liver fat content by measuring the attenuation coefficient (AC) on two-dimensional ultrasound images using the attenuation imaging (ATI) technique [16,17]. The attenuation coefficient corresponds to the change in US beam intensity with depth on US images. To obtain pure intensity change for the calculation of AC, influences such as gain control or focus-dependent US beam profile are removed from the observed intensity [16,17].

In veterinary medicine, a liver biopsy and MRI are rarely used diagnostic techniques and are sometimes contraindicated due to the need for general anesthesia and the risk of bleeding in suspected steatotic patients [18]. Therefore, the evaluation of liver parenchyma in dogs with hyperlipidemia is performed exclusively by B-mode ultrasound [19]. To date, only one study has been conducted in 2022 by Lee et al., which evaluated the applicability of the ATI procedure on healthy Beagle dogs [20].

The present study aimed to evaluate the applicability of ATI in dogs with hyperlipidemic status and ultrasonographic alterations suggestive for hepatic steatosis.

## 2. Materials and Methods

### 2.1. Selection and Description of Subjects

This prospective study includes dogs presented to the Veterinary Teaching Hospital “Mario Modenato” of the University of Pisa from January 2021 to December 2022. All procedures were approved by the Animal Research Ethics Board (OPBA protocol number 43/2020), and informed owner consent was obtained for each dog. The dogs included were then divided into two distinct populations: a group of non-hyperlipidemic patients as a control group (Group A) and a group of hyperlipidemic patients (Group B).

All dogs included in this study underwent a physical examination, complete blood count and serum biochemical examination, B-mode abdominal ultrasound and liver ultrasound attenuation imaging (ATI) procedure to a conclusive diagnosis. The blood count and biochemical examination had to be carried out within one week of the B-mode and ATI procedures. The dogs in Group A were represented by blood donor patients referred for periodical evaluation and were considered healthy.

The dogs in Group B were patients brought to our Veterinary Teaching Hospital for a specialist gastrointestinal examination and were divided into two subgroups: mild hyperlipidemic (Group B1), characterized by increased levels of either cholesterol or triglycerides, and moderate/severe hyperlipidemic (Group B2), characterized by increased levels of both cholesterol and triglycerides. Patients undergoing corticosteroid treatment or those with concurrent hepatic neoplastic disease were excluded. Dogs with a histological diagnosis of fibrosis or cirrhosis, portosystemic shunts, toxic or drug-induced hepatopathy, infectious agents or acute biliary disease were also excluded.

### 2.2. Data Recording and Analysis—Ultrasound and ATI Assessment

The following information was recorded for each patient: age, breed, sex, body weight (BW) and body condition score (BCS). Complete recent and remote clinical histories, with a particular focus on gastrointestinal signs and physical examination, were performed by an experienced clinician (V.M.) and recorded for each dog. Serum levels of cholesterol and triglycerides were investigated using an enzymatic colorimetric method (Liasys SAT 450 biochemical analyzer; Assel S.r.l., Rome, Italy). We considered increased cholesterol if over 280 mg/dL and increased triglycerides if over 90 mg/dL [21].

Transcutaneous abdominal ultrasound was performed by one experienced ultrasonographer (T.P.) using a Canon Aplio a CUS-AA000 (Canon Medical Systems Europe B.V., Zoetermeer, The Netherlands) with a 7.5 MHz microconvex and a 12 MHz linear probe. DICOM images and video files were stored in a workstation, evaluated using an open-source dedicated DICOM viewer (Horos v. 168 3.3.6, Horosproject.com) and reviewed by one author (T.P.). A single individual was used to retrospectively assess the images and videos to prevent interobserver variation from impacting the data. Dogs were examined without sedation or anesthesia.

At conventional B-mode ultrasound, liver echogenicity was evaluated with the following parameters to determine the degree of steatosis: (1) comparison between the echogenicity of the liver parenchyma and the right renal cortex; (2) poor or non-visualization of the intrahepatic portal vein wall, diaphragm and posterior part of the right hepatic lobe.

Therefore, hepatic echogenicity has been classified into four ultrasound patterns, as already validated in human medicine [13,14]. The description of the ultrasound grades of echogenicity is described and shown in Table 1 and Figure 1.

After the B-mode ultrasonography, an ATI examination was performed with a specific 3.5 MHz convex probe (PVT-375 BT). The dogs were positioned in left lateral recumbency to allow for visualization of the right hepatic lobe [20]. The measurement at this level would be more reliable and less influenced by the movements of the diaphragm, heart, lungs and stomach. The images were then obtained through an intercostal approach, positioning the probe at the level of one of the last 2–3 intercostal spaces, parallel to the ribs. During the ATI examination, the real-time grayscale B-mode was visible on the left side of the monitor, and the attenuation color map mode was shown on the right side, where we placed a field of view (FOV), varying in size depending on the patient size.

For scanning depth, we positioned the FOV at least 20 mm deep to prevent reverberation artifacts of the skin and liver capsule, liver capsule (dark orange area), and to obtain a homogeneous blue area within the FOV, avoiding large hepatic vessels and the low signal-to-noise ratio in the far field (dark blue area).

After finding the correct position, the screen was frozen, and within the FOV, a region of interest (ROI) (region of interest) was placed; the attenuation coefficient (AC), in the unit of decibels per centimeter per megahertz (dB/cm/MHz), appeared immediately. The coefficient of determination (R^2^) represents the reliability index of the measurements. Data quality was classified as excellent (R^2^ ≥ 0.9), good (0.9 > R^2^ ≥ 0.8) or poor (R^2^ < 0.8), and only data with an R^2^ value ≥ 0.9 were used (Figure 2). The examinations were repeated 5 times for each dog, and the mean, standard deviation, median, interquartile range (IQR) and ratio IQR/median were calculated.

### 2.3. Statistical Analysis

Statistical analyses were performed using commercial statistical software (GraphPad Prism v. 9.0, GraphPad Software Inc., San Diego, CA, USA) by a Ph.D. diagnostic imaging veterinarian with expertise in statistics (C.P.). The data distribution was assessed by using the Shapiro–Wilk test, and the data were expressed as the median and range or mean ± standard deviation according to their distribution. The difference between the ATI values of healthy and hyperlipidemic patients and between the different degrees of hyperlipidemia was evaluated using the unpaired *t*-test. The Spearman test was used to evaluate the possible correlation between the BCS and AC values in the hyperlipidemic patients. A *p*-value <0.05 was considered statistically significant.

## 3. Results

### 3.1. Study Population

A total of 53 patients were included in our study, with 21 healthy dogs in Group A (39.6%) and 32 dogs in Group B with an underlying condition of hyperlipidemia (60.4%).

Group A was composed of four females (three intact and one castrated) and seventeen males (eight intact and nine castrated). The most represented breed was Labrador Retriever (n = 19/21), with one American Pitbull Terrier and one mixed breed. The median age was 30 months (range 24–54 months). Ten patients out of 21 (47.6%) had a weight between 10 and 25 kg, and 11/21 dogs (52.4%) had a weight > 25 kg.

Group B was composed of 22 females (eleven intact and ten castrated) and 10 males (eight intact and two castrated). The different breeds included two Poodles, two Dachshunds, one German Dachshund, one Boxer, two Cavalier King Charles, two Cocker Spaniels, one Golden Retriever, two Jack Russel Terriers, one Labrador, one Maltese, one Pincher, one Schnauzer, one Scottish Terrier, one Setter Gordon, two Shih Tzu, one West Highland White Terrier, one Yorkshire and nine mixed breeds. The median age was 11 years (range 6–14.9 years). Sixteen patients out of 21 (50%) had a weight < 10 kg, 12/21 (37.5%) dogs had a weight between 10 and 25 kg, and 4/21 (12.5%) patients had a weight > 25 kg. Additionally, 15/21 (46.9%) dogs had an obesity condition with a BCS ≥ 6/9, 15/21 had normal weight (BCS = 4/9 or 5/9), and 2/21 were underweight (BCS = 3/6).

All dogs in Group B were divided into two other groups based on the degree of hyperlipidemia: Group B1 (mild hyperlipidemia; n = 15) and Group B2 (moderate/severe hyperlipidemia; n = 17) based on altered cholesterol and/or triglyceride value. No correlation was found between BCS and ATI values in the hyperlipidemic subjects (*p* = 0.47).

Thirteen of the 32 dogs with hyperlipidemia had an underlying endocrinopathy: hyperadrenocorticism (n = 4), diabetes mellitus (n = 2) and hypothyroidism (n = 7). For the other patients, the final diagnosis was cholecystopathy (n = 10), epatopathy (n = 5), triaditis (n = 2), primary hyperlipidemia (n = 1) and hyperlipidemia due to obesity (n = 1).

A quantitative assessment was performed for serum cholesterol and triglyceride values for the patients included in Group B for both mild and moderate/severe hyperlipidemic dogs. The median and range of only the increased value of cholesterol (n = 12) and triglycerides (n = 3) were calculated in the dogs with mild hyperlipidemia, while in the dogs with moderate/severe hyperlipidemia (n = 17), both parameters were measured in all dogs (Table 2).

### 3.2. Imaging and Statistical Analysis

Among the dogs belonging to Group B, ultrasound B-mode evaluation of the liver parenchyma showed 2/32 (6.25%) patients with grade 0 echogenicity, 16/32 (50%) with grade 1, 12/32 (37.5%) with grade 2 and 2/32 (6.25%) with grade 3 (Table 3).

The ATI evaluation took no more than 5 min in all patients; it was a well-accepted procedure by all our patients and did not create any discomfort conditions.

The AC values between Group A and Group B showed a statistically significant difference (*p* = 0.01), with higher values in the hyperlipidemic dogs. The values expressed as mean ± standard deviation are shown in Table 4 and Figure 3. In Figure 4, there are two examples of measurement in one patient of Group A and one of Group B. (Figure 3).

No significant differences were observed in the ATI values between the mild hyperlipidemic and moderate/severe hyperlipidemic dogs (*p* = 0.44); the values are shown in Table 5 and Figure 5.

In Group B, it was not possible to evaluate a possible statistical difference in the AC values between the various degrees of echogenicity of the liver parenchyma because grade 0 and grade 3 included a small number of patients (Table 6).

However, it is possible to notice an increasing value of attenuation from grade 0 to grade 3.

## 4. Discussion

Our study evaluated the use of ATI for the non-invasive assessment of suspected liver steatosis in a population of dogs with a hyperlipidemic condition; ATI proved to be a minimally invasive technique, easy to use and perform, and, if the patient is cooperative, also rapid.

In human studies, the examination protocol requires the patient to be in apnea [16,22,23]. In our study, the ATI values were all reliable (R^2^ value ≥ 0.9) regardless of the patient’s breathing, as reported in a previous study by Lee [20]. It is certainly possible that excessive movement in the case of a tachypneic patient could influence the results or make the exam impossible.

The first important result from our study is that the mean AC value was significantly higher in hyperlipidemic patients than in healthy ones. This finding agrees with what has been recently demonstrated in human medicine [16,22,23].

The mean AC value in the healthy dogs was higher (0.81 dB/cm/MHz) than the value reported in the only study carried out in veterinary medicine, in which the AC value was 0.64 dB/cm/MHz at 10–20 mm depth and 0.54 dB/cm/MHz at 20–30 mm depth [20]. This difference could be related to the small number of subjects examined (10 dogs) that were of a single breed (Beagle), all males of similar weight; instead, in our study, we included a larger number of patients with a wider weight range and larger range of examination depth. Moreover, we used a different model of convex probes.

Despite the significant difference in the AC values of Groups A and B, an overlapping range was identified, and it was not possible to determine a significant difference in the AC values as a function of the degrees of echogenicity of the liver parenchyma or of the values of hyperlipidemia. There may be some explanations. In the first place, the overlap occurs mainly for patients with grades 1 and 2 of echogenicity, which are intermediate grades, while it is not present for grades 0 and 3, which, however, were poorly represented. We can deduce that ATI could probably differentiate severe steatosis stages but not mild/moderate states.

Furthermore, the clinical classification of the severity of the hyperlipidemic state of patients based only on the value of cholesterol and triglycerides may not have been sufficient, not having considered other variables such as the time of onset or the final diagnosis. In addition, it was found that, in the early stages of hyperlipidemia, slightly higher biochemical values than the normal range associated with grade 0 of liver echogenicity can be observed.

Therefore, blood values may be more sensitive in identifying hyperlipidemia than ultrasound evaluation; in human medicine, it is known that the diagnostic accuracy of evaluating parenchymal echogenicity is not high when fat accumulation in the liver is low [17,24,25]. Therefore, ultrasound examination in mild hyperlipidemia may show no change in both the echogenicity of the liver parenchyma and AC values in dogs. However, from the results obtained, it is possible to detect a progressive increase in AC values in the groups of patients classified from grade 0 to 3 of hepatic echogenicity.

No statistically significant difference was found in the AC values between the patients with mild hyperlipemia (Group B1) and moderate/severe (Group B2) hyperlipidemia. This statistical result could have been influenced by the classification criteria for hyperlipidemia and by the small number of patients included in Groups B1 and B2.

To critically evaluate the data obtained in our study, it is appropriate to consider that there are some variables that can influence the results of the ATI exam. Studies in human medicine have shown that body mass index, obesity and skin–liver–capsule distance can influence the AC value, overestimating it [16,22,23,26]. To date, in veterinary medicine, there are no studies that have investigated the influence of these variables in the quantification of AC values; in our study, no correlation was found between BCS and ATI value in hyperlipidemic patients (*p* = 0.47).

The main limitation of our study is the lack of a biopsy or fine needle biopsy to confirm the presence of hepatic steatosis in the included dogs. This is a preliminary study aimed to evaluate the applicability of the ATI technique, and further studies are needed to validate this method, referring to cytologic/histopathological features. Furthermore, it should be considered that, in human medicine, only PDFF MRI is currently the most accurate quantitative imaging biomarker of steatosis [12]. A second limitation of this study is that the group of pathological dogs was heterogeneous in breed, age and weight. In veterinary medicine, we do not know the influence of these variables on the determination of the AC, while it is known that, in humans, weight can influence the outcome of the ATI study.

Lastly, we did not evaluate potential interobserver variability, even if the method clearly limits measurement errors.

## 5. Conclusions

In conclusion, ATI represents a promising technique in veterinary medicine for the non-invasive identification and quantification of suspected hepatic steatosis. This study demonstrated that the AC values in dogs with a hyperlipidemic condition are significantly higher than in healthy patients. Attenuation imaging could be a useful diagnostic tool to be combined with the clinic, both for an initial quantification of steatosis in the case of diagnostic suspicion and for patient monitoring. Further studies will be needed to expand the population, to have a greater number of dogs, especially belonging to groups 2 and 3 of hepatic ultrasound alteration referable to steatosis, to verify if ATI is able to differentiate different severities of liver disease. Another objective is to make temporal assessments of patients to verify whether a clinical and laboratory variation corresponds to an ATI variation.

## Figures and Tables

**Figure 1 vetsci-11-00454-f001:**
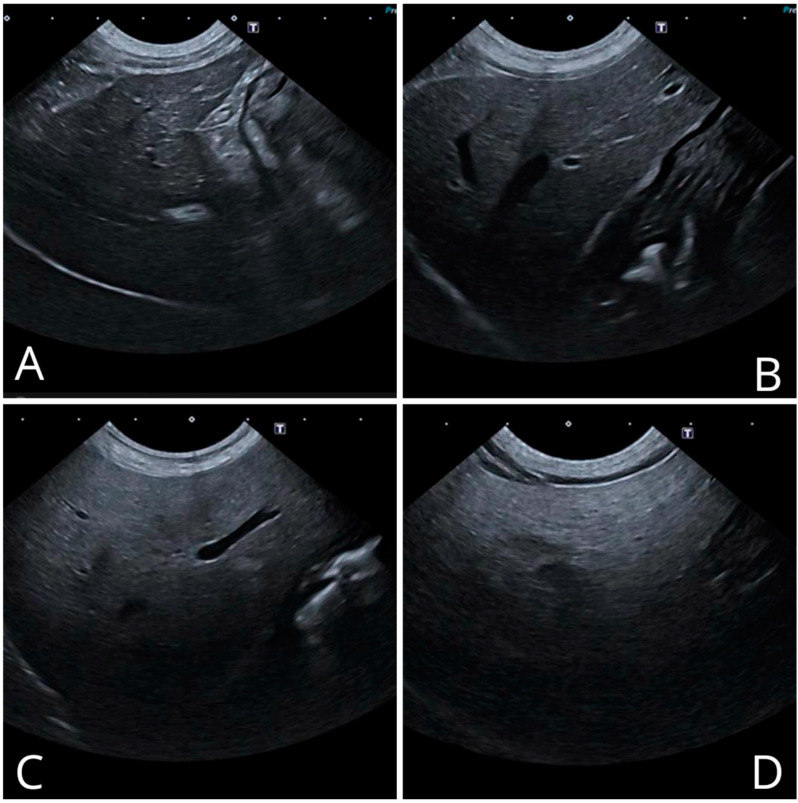
Ultrasound images of the hepatic parenchyma showing the different grades of echogenicity: (**A**) grade 0, (**B**) grade 1, (**C**) grade 2, (**D**) grade 3.

**Figure 2 vetsci-11-00454-f002:**
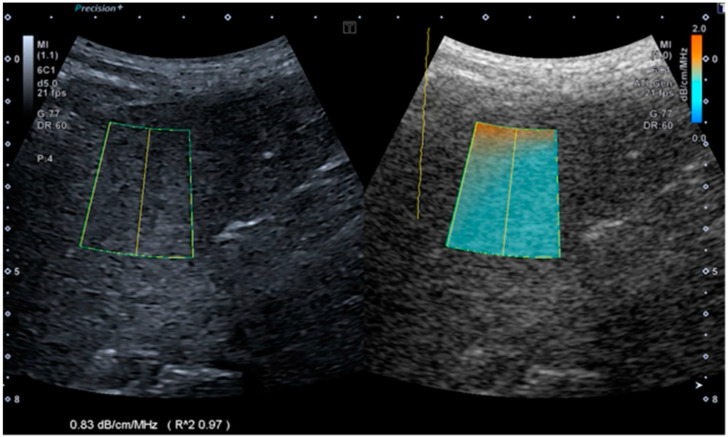
Image obtained during ultrasonographic attenuation imaging of the hepatic parenchyma in a dog from our study. The real-time B-mode image is on the left side, and the attenuation color map is on the right side. The field of view (FOV) and region of interest have the same size. There is a homogeneous blue area inside the FOV. In this dog, the calculated attenuation coefficient is 0.83 dB/cm/MHz, and the R^2^ value is 0.97, indicating a reliable measurement result.

**Figure 3 vetsci-11-00454-f003:**
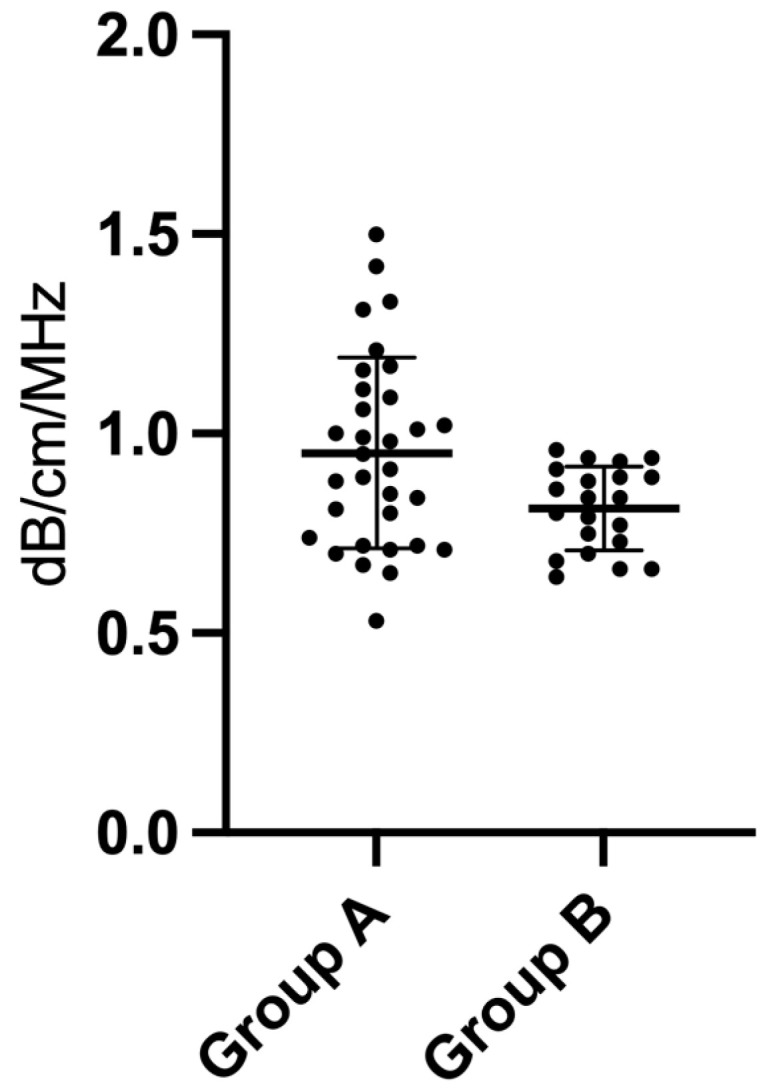
Attenuation coefficient values expressed in mean ± standard deviation for Group A and Group B.

**Figure 4 vetsci-11-00454-f004:**
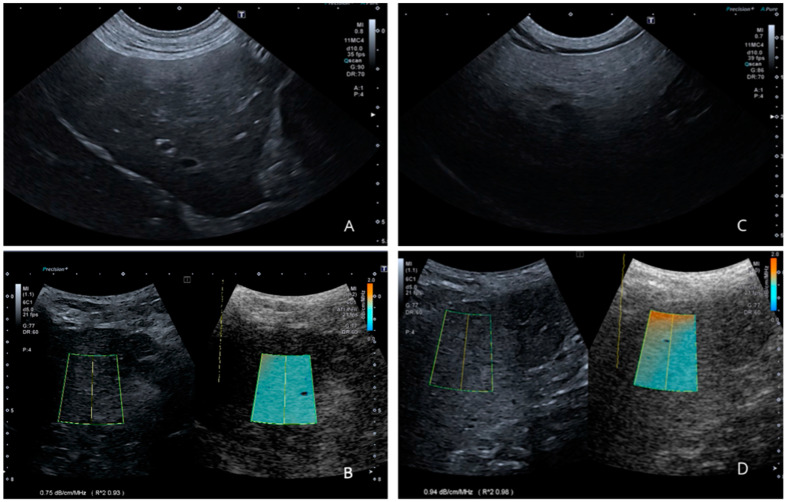
(**A**,**B**): female, 3.6 y/o of Group A. (**A**) shows the echogenicity of the liver parenchyma with a grade 0. (**B**) shows how the split screen looks when performing the attenuation imaging (ATI) method. In this case, the attenuation coefficient (AC) value is 0.83 dB/cm/MHz, with optimal reliability indicated by the R^2^ value > 0.90. (**C**,**D**): female, 10.4 y/o of Group B. (**C**) shows the echogenicity of the liver parenchyma with a grade 3. (**D**) shows how the split screen looks when performing the ATI method. In this case, the AC value is 0.94 dB/cm/MHz, with optimal reliability indicated by the R^2^ value > 0.90.

**Figure 5 vetsci-11-00454-f005:**
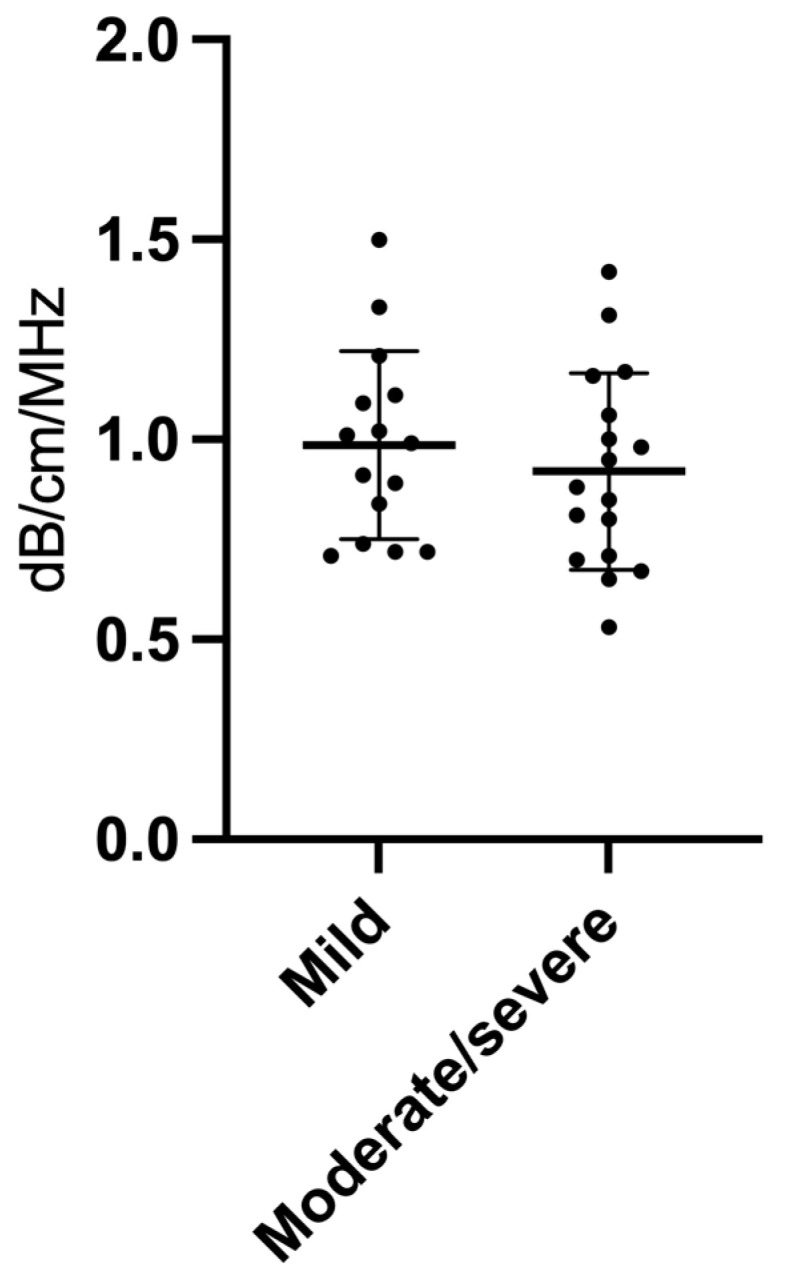
Attenuation coefficient values expressed in mean ± standard deviation for patients with mild hyperlipidemia and with moderate/severe hyperlipidemia.

**Table 1 vetsci-11-00454-t001:** Grades of hepatic echogenicity based on the ultrasound appearance.

	Hepatic Echogenicity Degree	Ultrasound Appearance
Grade 0	Normal	Homogeneous ecostructure, slightly increased echogenicity compared with normal renal cortical
Grade 1	Mild	Diffuse slight increase in liver echogenicity with normal visualization of the diaphragm and portal vein wall
Grade 2	Moderate	Moderate diffuse increase in liver echogenicity with reduced definition of the portal vein wall and diaphragm
Grade 3	Severe	Severe increase in liver echogenicity with poor or no visualization of the portal vein wall, diaphragm and posterior portion of the right hepatic lobe

**Table 2 vetsci-11-00454-t002:** Cholesterol and/or triglyceride values (mg/dL) expressed as median and range in the bracket, calculated for dogs with mild and moderate/severe hyperlipidemia. Normal values: cholesterol (120–280) and triglycerides (25–90).

Mild (n = 15)		Moderate/Severe (n = 17)	
Cholesterol (n = 12)	Triglycerides (n = 3)	Cholesterol	Triglycerides
359 (298–624)	176 (99–1034)	418 (295–789)	240 (93–790)

**Table 3 vetsci-11-00454-t003:** Classification of hyperlipidemic dogs based on the degree of echogenicity and mild or moderate/severe hyperlipidemia.

Grade of Echogenicity	Number of Patients (n = 32)	Grade of Hyperlipidemia	
		Mild	Moderate/Severe
Grade 0	2 (6.25%)	2 (6.25%)	0
Grade 1	16 (50%)	8 (25%)	8 (25%)
Grade 2	12 (37.2%)	5 (15.6%)	7 (21.8%)
Grade 3	2 (6.25%)	0	2 (6.25%)

**Table 4 vetsci-11-00454-t004:** Mean ± standard deviation and ranges of the attenuation coefficient (AC) values in patients of Group A and Group B.

	AC (dB/cm/MHz)	Range
Group A (n = 21)	0.81 ± 0.10	0.64–0.96
Group B (n = 32)	0.95 ± 0.23	0.53–1.50
	*p* = 0.01	

**Table 5 vetsci-11-00454-t005:** Mean ± standard deviation and ranges of the attenuation coefficient (AC) values (dB/cm/MHz) in patients with mild and moderate/severe hyperlipidemia.

Hyperlipidemia	AC (dB/cm/MHz)	Range
Mild (n = 15)	0.98 ± 0.23	0.71–1.50
Moderate/severe (n = 17)	0.92 ± 0.24	0.53–1.42
	*p* = 0.44	

**Table 6 vetsci-11-00454-t006:** Mean ± standard deviation and ranges of the attenuation coefficient (AC) values (dB/cm/MH) in patients of Group B with different degrees of echogenicity of the hepatic parenchyma.

Echogenicity	AC (dB/cm/MHz)	Range
*Grade 0 (n = 2)*	0.80 ± 0.08	0.83–0.95
*Grade 1 (n = 16)*	0.95 ± 0.27	0.62–1.54
*Grade 2 (n = 12)*	0.93 ± 0.26	0.49–1.54
*Grade 3 (n = 2)*	1.10 ± 0.3	0.89–1.32

## Data Availability

The data presented in this study are available on request from the corresponding author.

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
