# Peer review of "Evaluation of Liver Parenchyma in Dogs with Hyperlipidemia Using Ultrasound Attenuation Imaging (ATI)"

_vetsci, 2024, doi:10.3390/vetsci11100454_

Round 1

Reviewer 1 Report

Comments and Suggestions for Authors

The manuscript explores the use of ultrasound attenuation imaging (ATI) to diagnose hepatic steatosis in dogs with hyperlipidemia, a condition characterized by elevated levels of lipids in the blood. The study aims to determine the applicability of ATI in veterinary medicine and its correlation with traditional B-mode ultrasound.

Based on a critical assessment, here are several weaknesses and areas of concern in the article.

1.     How consistent are the ATI measurements when performed by different operators? Please Include a discussion on potential inter-operator variability in the ATI procedure and any measures taken to minimize this variability.

2.     Were there any standardized protocols or guidelines followed for performing ATI measurements across different dogs?

3.     What are the specific clinical implications of different ATI values for diagnosing and managing hepatic steatosis in dogs?

4.     Is there any longitudinal data available to assess the changes in ATI values over time in hyperlipidemic dogs undergoing treatment?

5.     What are the sensitivity and specificity of ATI in detecting various grades of hepatic steatosis in dogs?

6.     What are the future research directions for validating and improving the use of ATI in veterinary medicine?

Comments on the Quality of English Language

Moderate editing of English language required.

Author Response

The authors (AU) would like to thank the reviewers for taking the time to review this manuscript. We found the comments appropriate and very useful. The changes to the manuscript have characters in yellow color for the first Reviewer. The replies to the reviewers' comments are reported below. We have also identified and corrected several minor errors in the text

1. How consistent are the ATI measurements when performed by different operators? Please Include a discussion on potential inter-operator variability in the ATI procedure and any measures taken to minimize this variability.

AU: Thank you for your question. We have added a statement at the end of the introduction to clarify the study's purpose, which was to evaluate whether the ATI method could detect variations in liver attenuation values in hyperlipidemic subjects compared to healthy controls, and to assess whether, similar to human medicine, it could differentiate between varying degrees of severity. Our study did not include an analysis of interobserver variability because this has already been extensively studied in human medicine, with operators of varying levels of experience (Jae Seok Bae, Dong Ho Lee, Jae Young Lee, Haeryoung Kim, Su Jong Yu, Jeong-Hoon Lee, Eun Ju Cho, Yun Bin Lee, Joon Koo Han, Byung Ihn Choi. "Assessment of hepatic steatosis using attenuation imaging: a quantitative, easy-to-perform ultrasound technique." Eur Radiol. 2019 Dec;29(12):6499-6507), as well as across different ultrasound platforms (Han A, Zhang YN, Boehringer AS et al. "Inter-platform reproducibility of ultrasonic attenuation and backscatter coefficients in assessing NAFLD." Eur Radiol. 2019. https://doi.org/10.1007/s00330-019-06035-9). Both studies showed no significant differences in results. This is because the ATI method provides a numerical accuracy index through its detection software, below which measurements are not accepted. In the only veterinary study by Lee, the repeatability of the measurements, even when performed by a single operator, was deemed reliable, both at different detection sites and at scanning depths of 10-20 mm. However, we acknowledge that we do not possess this data and consider it a limitation of our study.

2. Were there any standardized protocols or guidelines followed for performing ATI measurements across different dogs?

AU: As mentioned in the previous response, the ATI method minimizes subjectivity in measurements due to the coefficient of determination (R2), which serves as an index of measurement reliability. This value is displayed on the monitor during measurements, indicating their reliability. We consistently recorded attenuation coefficient (AC) measurements only when the R² value was ≥ 0.9, a threshold considered excellent (see lines 167-172). Additionally, measurements were performed on dogs positioned in left lateral recumbency to enable visualization of the right hepatic lobe, a procedure that is feasible across all breeds and thoracic conformations. Moreover, we maintained a consistent measurement depth by placing the field of view (FOV) at least 2 cm from the skin (line 163), which is naturally easier in dolichomorphic dogs with a low body condition score (BCS). To date, there are no other studies of this kind in the literature, as the only existing study on ATI in dogs is by Lee et al. 2022, which evaluated the feasibility of hepatic parenchyma ATI in 10 healthy beagles. In our study, no correlation was found between BCS and ATI values in hyperlipidemic subjects, which contrasts with findings in humans.

3. What are the specific clinical implications of different ATI values for diagnosing and managing hepatic steatosis in dogs?

AU: Dear reviewer, thank you for the interesting point. We believe that ATI (once validated) may be an useful tool (easy applicability, relatively cheap) to monitor the evolution, in terms of follow-up, of hepatic injury. I.e. endocrinopathic patients.

4. Is there any longitudinal data available to assess the changes in ATI values over time in hyperlipidemic dogs undergoing treatment?

AU: Our study builds upon the work of Lee, who standardized the ATI method for evaluating the liver in healthy dogs. Our primary objective was to assess whether this method could serve as a non-invasive, quantitative tool for evaluating the liver in hyperlipidemic dogs without the need for anesthesia. It was challenging to include a large sample size due to strict inclusion criteria and the liver echogenicity classification system we adopted from human literature (Ferraioli, G.; Berzigotti, A.; Barr, R.G.; Choi, B.I.; Dong, Y. et al. "Quantification of Liver Fat Content with Ultrasound: A WFUMB Position Paper." Ultrasound Med Biol 2021, 47, 2803-2820).

However, this study marks the beginning of a new research direction with several future goals. One of these is to expand the sample size, particularly to include more dogs in groups 2 and 3 of hepatic ultrasound echogenicity, consistent with suspected steatosis, and to assess further ATI's ability to differentiate between these classes. Another goal is to perform longitudinal assessments to determine whether changes in clinical and laboratory parameters correlate with changes in ATI values. These future research directions are outlined at the end of the discussion (lines 335-360).

5. What are the sensitivity and specificity of ATI in detecting various grades of hepatic steatosis in dogs?

AU: After the observation from Reviewer 2, we decided to remove the term "steatosis" in the absence of histological confirmation and instead refer only to liver ultrasound findings in hyperlipidemic patients.

Our study demonstrates that the ATI method can be applied to this patient group, showing that hyperlipidemic dogs have significantly higher hepatic ATI values compared to healthy dogs. However, when we categorize patients into the four ultrasound classes based on human literature, the sample size is too small to achieve statistically significant attenuation coefficient values. Therefore, future research will focus on increasing the sample size to determine whether ATI can, as in humans, differentiate between varying severities of liver disease.

6. What are the future research directions for validating and improving the use of ATI in veterinary medicine?

AU: As mentioned in response to question 4, our next objective will be to increase the number of hyperlipidemic patients in order to evaluate whether the attenuation coefficient from ATI provides good diagnostic performance in detecting varying degrees of suspected hepatic steatosis. Additionally, we aim to assess whether ATI could serve as a promising tool for screening and monitoring hepatic alterations.

Reviewer 2 Report

Comments and Suggestions for Authors

Show the results more clearly in the summary and in the Simple Summary.

What were the criteria used to define the degree of hyperlipidemia in group B?

What were the criteria used to define the degrees of steatosis? Are they validated for the species studied? If so, include them as a reference in the table.

Improve the quality of the figures in Figure 1.

What calculations were used in the statistical planning to define the sample size, especially for group B?

Group B was subdivided into different degrees of hyperlipidemia and steatosis, but all of them were combined to compare with Group A. This may have generated a biased result, since the degrees of hyperlipidemia and steatosis were different. The number of these sample groups was also very different, which would make it difficult to compare the different groups. All of this generated biased results that cannot be used for conclusions as was done.

In the conclusion, it would not be prudent to conclude a trend if there is no statistical evidence.

Author Response

The authors (AU) would like to thank the reviewers for taking the time to review this manuscript. We found the comments appropriate and very useful. The changes to the manuscript have characters in  green color for the second Reviewer. The replies to the reviewers' comments are reported below. We have also identified and corrected several minor errors in the text.

1. Show the results more clearly in the summary and in the Simple Summary.

AU: Thanks for your comment. We modified the Results part in the Simple summary (lines 17-20) and in the Abstract (lines 29-32).

2. What were the criteria used to define the degree of hyperlipidemia in group B?

AU: Dear reviewer, thank you for your observation. We added a pertinent reference referring to the classification used.

3. What were the criteria used to define the degrees of steatosis? Are they validated for the species studied? If so, include them as a reference in the table.

AU: Thank you for your comment. To define the degree of suspected steatosis ultrasonographically, we used a B-mode classification based on liver parenchymal echogenicity, a method that has been validated in human medicine. Currently, no validated methods for classifying liver echogenicity in steatosis are available for dogs. We have revised a sentence in the Results section to clarify this (lines 141-143).

4. Improve the quality of the figures in Figure 1.

AU: Thanks for your comment. We tried to brighten the image as a whole to allow more detail of the liver parenchyma.

5. What calculations were used in the statistical planning to define the sample size, especially for group B?

AU: Thank you for your observation. A post-hoc power analysis, using a t-test to compare two independent means between unpaired groups, was conducted to estimate the sample size for Groups A and B, as well as Groups B1 and B2. This analysis was performed using G*Power (Ver. 3.1, Heinrich-Heine-Universität). The results showed a power of 87% for Groups A and B, and 17% for Groups B1 and B2. Based on these findings, the statistically significant difference in hepatic attenuation values between Groups A and B can be considered reliable. However, the lack of a statistically significant difference between Groups B1 and B2 may be due to the limited number of patients in these subgroups.

The research study was approved by the ethics committee, and we were operating within a time constraint for patient recruitment, during which we included as many patients as possible. One of the key goals for future studies is to increase the number of patients, particularly in the hyperlipidemic group and its liver echogenicity subgroups. If you feel it is necessary, we are happy to include this statistical analysis in the manuscript. We have also added a sentence to the discussion section (lines 327-330).

6. Group B was subdivided into different degrees of hyperlipidemia and steatosis, but all of them were combined to compare with Group A. This may have generated a biased result, since the degrees of hyperlipidemia and steatosis were different. The number of these sample groups was also very different, which would make it difficult to compare the different groups. All of this generated biased results that cannot be used for conclusions as was done. In the conclusion, it would not be prudent to conclude a trend if there is no statistical evidence.

AU: Thanks for your comment. We chose not to compare Group A with Groups B1 and B2 considering that there was no significant difference between liver attenuation values between Groups B1 and B2. The absence of significant difference in this case could be secondary to the small number of subjects included in the two groups, or to the classification of hyperlipidemia which in this case took into account only the presence of an increase in cholesterol and/or triglyceride values. We did not perform a statistical analysis to evaluate the correlations between degrees of liver echogenicity, attenuation values and degree of hyperlipidemia, considering the small number of patients included in the groups to be compared. Consequently, we have removed from the text the objective of assessing the correlation between liver attenuation coefficients, echogenicity degrees, and hyperlipidemia.

Reviewer 3 Report

Comments and Suggestions for Authors

In the present study, the authors assess the clinical applicability of the ultrasound attenuation imaging (ATI) method for evaluating hepatic parenchyma in healthy dogs and those with hyperlipidemia. Overall, the article is well-structured, and given the lack of related literature in veterinary medicine, it could shed light on the utility of this method in canine patients with liver pathology. However, the following minor modifications or changes are recommended:

- Lines 202-221: Much of the information in this paragraph should be included in the materials and methods section rather than in the results section. For instance, the breeds or the number of individuals in each group are not results of the study but rather part of the methodology used.

-In the materials and methods section, it would be beneficial to illustrate the placement of the probe during the procedure and the obtained scans more graphically. Schematic diagrams could be employed. This would be of great importance for facilitating the broader adoption and reproducibility of the technique.

- In the results section, it would be advisable to include graphs alongside the tables to represent the various outcomes obtained. The presentation of results is one of the most critical parts of the article, and in this case, it is the weakest in terms of format (furthermore, as mentioned earlier, some of the information in this section belongs in the materials and methods section).

- Do the authors consider it feasible to perform an adequate ultrasonographic evaluation of an organ like the liver, given its location, in large-breed dogs WITHOUT SEDATION?

Beyond the comments mentioned above, the article presents a series of significant limitations, which the authors have addressed in the discussion section (along with the introduction, the the most well-developed parts of the document). For instance, it would be important to consider the technical factors that may affect the reproducibility of the technique and to expand on this section (lines 326-331). Additionally, the authors mention the absence of other techniques such as MRI, CT, or biopsy. Indeed, it would be crucial, especially for validating and standardizing the method, to use these techniques to confirm the results. For example, hepatic FNAs could be employed in cases where there are economic constraints or specific risks to the patient to assess the correlation between the ultrasound image and actual hepatic abnormalities. Finally, and very notably, with regard to the robustness of the data obtained, the heterogeneity of the different groups (primarily breed and size) is concerning, especially given the prospective nature of the study.

Taking all of this into account, and considering the scarcity of data on this topic in veterinary medicine, the article could be considered for publication, provided that these modifications and limitations are addressed, and laying the groundwork for more extensive studies with better group structuring.

Author Response

The authors (AU) would like to thank the reviewers for taking the time to review this manuscript. We found the comments appropriate and very useful. The changes to the manuscript have characters in  light blue for the third Reviewer. The replies to the reviewers' comments are reported below. We have also identified and corrected several minor errors in the text.

1. Lines 202-221: Much of the information in this paragraph should be included in the materials and methods section rather than in the results section. For instance, the breeds or the number of individuals in each group are not results of the study but rather part of the methodology used.

AU: Thank you for your comment. Our study was prospective, with the primary objective of evaluating the clinical applicability of the ATI method, given that Lee et al. had already assessed its feasibility. To maximize the number of patients, we included both healthy and hyperlipidemic patients without specific criteria related to signalment. As a result, no a priori inclusion criteria for age, weight, or breed were established. For this reason, this information has been included in the results section. Below are some prospective studies in which such data have been included in the results:

  1. S D Kemp, D L Panciera, M M Larson, G K Saunders, S R Werre. A comparison of hepatic sonographic features and histopathologic diagnosis in canine liver disease: 138 cases. J Vet Intern Med. 2013 Jul-Aug;27(4):806-13. doi: 10.1111/jvim.12091. Epub 2013 May 6.
  2. Jae Seok Bae, Dong Ho Lee, Jae Young Lee, Haeryoung Kim, Su Jong Yu, Jeong-Hoon Lee, Eun Ju Cho, Yun Bin Lee , Joon Koo Han. Assessment of hepatic steatosis by using attenuation imaging: a quantitative, easy-to-perform ultrasound technique. Eur Radiol. 2019 Dec;29(12):6499-6507. doi: 10.1007/s00330-019-06272-y. Epub 2019 Jun 7.
  3. Caterina Puccinelli, Tina Pelligra, Angela Briganti, Simonetta Citi. Two-dimensional shear wave elastography of liver in healthy dogs: anaesthesia as a source of variability. Int J Vet Sci Med. 2022; 10(1): 46–51. 2022 May 23. doi: 10.1080/23144599.2022.2073138

2. In the materials and methods section, it would be beneficial to illustrate the placement of the probe during the procedure and the obtained scans more graphically. Schematic diagrams could be employed. This would be of great importance for facilitating the broader adoption and reproducibility of the technique.

AU: Thank you for your observation. The positioning of the probe has been extensively described in Lee et al. (2022) concerning the ATI method, as well as in studies on 2D-SWE for liver parenchyma evaluation, where the probe positioning is identical. The intercostal approach for visualizing the right hepatic lobe makes the ATI examination straightforward and easily repeatable. For this reason, we did not initially include a schematic diagram to illustrate this approach. However, if you believe it would be beneficial, we would be happy to add it.

3. In the results section, it would be advisable to include graphs alongside the tables to represent the various outcomes obtained. The presentation of results is one of the most critical parts of the article, and in this case, it is the weakest in terms of format (furthermore, as mentioned earlier, some of the information in this section belongs in the materials and methods section). 

AU: Thank you for your comment. We have included two graphs (Figures 3 and 5) to visually represent the attenuation coefficient values in the comparison between Group A and Group B, as well as between patients with mild hyperlipidemia and those with moderate to severe hyperlipidemia.

4. Do the authors consider it feasible to perform an adequate ultrasonographic evaluation of an organ like the liver, given its location, in large-breed dogs WITHOUT SEDATION?

AU: As previously addressed in response to the second question from Reviewer 1, the probe was applied to the right hypochondrium with the dog in left lateral recumbency. The liver parenchyma was consistently visualized beneath the skin, even in 15 out of 21 dogs with a body condition score of ≥6/9. A potential significant drawback could have been polypnea, but this was not encountered, possibly due to the advanced age of our patients (average age 11 years). Similar findings have been reported in other studies utilizing comparable approaches for liver examination (e.g. Puccinelli et al., "Two-dimensional shear wave elastography of liver in healthy dogs: anesthesia as a source of variability," Int J Vet Sci Med. 2022; 10(1):46–51, doi: 10.1080/23144599.2022.2073138). Likewise, Toom et al. reported only one patient out of 21 being excluded due to poor cooperation, and this was in younger subjects (Toom et al., "Shear wave elastography measurements in dogs treated surgically for congenital extrahepatic portosystemic shunts," Front Vet Sci. 2022 Sep 26;9:991148, doi: 10.3389/fvets.2022.991148).

5. Beyond the comments mentioned above, the article presents a series of significant limitations, which the authors have addressed in the discussion section (along with the introduction, the the most well-developed parts of the document). For instance, it would be important to consider the technical factors that may affect the reproducibility of the technique and to expand on this section (lines 326-331). Additionally, the authors mention the absence of other techniques such as MRI, CT, or biopsy. Indeed, it would be crucial, especially for validating and standardizing the method, to use these techniques to confirm the results. For example, hepatic FNAs could be employed in cases where there are economic constraints or specific risks to the patient to assess the correlation between the ultrasound image and actual hepatic abnormalities. Finally, and very notably, with regard to the robustness of the data obtained, the heterogeneity of the different groups (primarily breed and size) is concerning, especially given the prospective nature of the study.

AU: Dear reviewer, we understand your concern and totally relate. We modified the aim and the limitations section accordingly, emphatizing the preliminary nature of the study and the need for further studies aimed to validate ATI technique in relation to cyto/hytopathology. The authors wonder whether they have addressed the Reviewer’s questions satisfactorily. 

Round 2

Reviewer 1 Report

Comments and Suggestions for Authors

The authors have addressed my concerns.